# Specializing Large Models for Oracle Bone Script Interpretation via Agent-Driven Multimodal Knowledge Augmentation

## Abstract

Deciphering oracle bone script, which originated over 3,000 years ago and represents the earliest known mature writing system in China, is fascinating and highly challenging. Vision language models (VLMs) offer strong capabilities in perception, understanding, and reasoning, presenting opportunities for cross-disciplinary research. However, their lack of domain-specific knowledge often results in suboptimal performance. Existing approaches largely frame decipherment as an image recognition task, overlooking the hieroglyphic nature of oracle bone script and the structural and semantic information embedded in its component-based design. To address these challenges, we propose an agent-driven multimodal retrieval-augmented generation (RAG) framework that enables large models to act as domain experts for oracle bone research. We also introduce OB-Radix, a component-level oracle bone script dataset annotated by domain experts, which provides essential structural and semantic information absent from prior datasets. Furthermore, guided by expert knowledge, we design three benchmark tasks to systematically evaluate the ability of VLMs in oracle bone decipherment. Experimental results demonstrate that our framework produces more detailed and accurate interpretations than baseline methods. Beyond oracle bone script, our framework establishes a methodological foundation for applying large models to the decipherment of other logographic writing systems.

## 1 Introduction

Oracle Bone Script (OBS), the earliest known mature writing system in China, holds significant historical and cultural value. Among the more than 4,500 OBS characters, only about one-third have been successfully deciphered, leaving a vast number of glyphs with untapped interpretive potential (Li et al., 2024). Each undeciphered character may provide critical insights into ancient institutions, technologies, and beliefs. However, the fragmented and stylized nature of OBS inscriptions, combined with the necessity of deep paleographic and contextual expertise, makes decipherment particularly challenging.

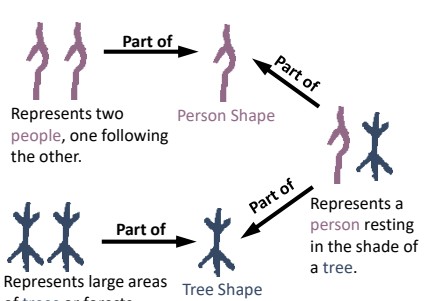

Figure 1: Oracle Bone Script (OBS), a pictographic writing system of semantic components.

In recent years, artificial intelligence has been increasingly applied to OBS interpretation (Fu et al., 2022; Wang et al., 2024a; Guan et al., 2024b; Jiang et al., 2023). Yet, most existing approaches rely primarily on a single modality (e.g., image recognition) while neglecting the structural, semantic, and contextual information intrinsic to the script. This narrow perspective not only risks information loss but also introduces interpretative biases, particularly in the absence of effective integration with domain-specific paleographic knowledge.

To address this multimodal challenge, we employ advanced VLMs. Prior work has shown that such models exhibit strong image–text understanding capabilities (Liu et al., 2023; Caffagni et al., 2024).

Figure 2: Comparison of our proposed framework and baselines. We design an agentic RAG framework to integrate component-level knowledge for structured semantic augmentation of OBS.

However, their competence in fine-grained perception and reasoning remains limited, and the lack of domain expertise leads to suboptimal performance in specialized tasks (Chen et al., 2025; Ye, 2024).

Specifically, OBS is a mature form of pictographic writing, structurally composed of multiple components, each carrying distinct semantic significance and often reused across different characters (Figure 1). Capturing this component-based structure is essential for accurate interpretation. To this end, we enhance VLMs with an Agentic Retrieval-Augmented Generation (Agentic RAG) framework that integrates component-level knowledge for structured semantic augmentation (Figure 2). It is designed to serve three tasks: component retrieval, component relationship inference, and OBS interpretation. Moreover, annotated data for oracle bone script remains scarce. Existing datasets, such as HUST-OBC and EVOBC (Wang et al., 2024b; Guan et al., 2024a), as well as other character-level datasets including OBC306 (Huang et al., 2019), Oracle-50k (Han et al., 2020), and HWOBC (Li et al., 2020), contain only complete character images without any component-level annotation. Although recent works have explored structure-related information, they do not provide true component-level data. Hu et al. (Hu et al., 2024) introduce the OBI Component-20 dataset for component-level retrieval, but its "components" are automatically extracted visual fragments rather than expert-verified semantic components. OracleSage (Jiang et al., 2024) incorporates structural knowledge through text-based decomposition, yet it does not supply image-level component segmentation. To address this gap, we introduce OB-Radix, a component-level oracle bone script dataset constructed under the guidance of professional archaeologists. Unlike prior datasets, OB-Radix provides expert-curated component images, semantic interpretations, and hierarchical structural relations, enabling fine-grained analysis of oracle bone script beyond the character level. In summary, our contributions are as follows:

- We propose a multimodal framework that integrates component-level visual cues with an Agentic RAG module, empowering VLMs to perform OBS tasks at an expert level.
- We construct OB-Radix, a component-level oracle bone script dataset, and build a knowledge graph that captures relationships among components, characters, and their semantic explanations, providing essential structured knowledge.
- We design comprehensive evaluations to ensure both the accuracy and interpretability of our approach. Results show that our framework achieves expert-level quality, with the multi-agent extension delivering strong semantic grounding and alignment with domain expert reasoning.

## 2 RELATED WORK

**Deciphering of Oracle Bone Script.** Existing research relies on a single image morphology model to explore AI reading paths. Guan et al. (2024b) employs a diffusion approach to map oracle bone inscription images to modern Chinese characters, while Qiao et al. (2024) leverages image generation to provide visual interpretive guidance. However, the former lacks integration of textual semantics, and the latter results in incomplete understanding due to the absence of textual guidance. Other studies applied diverse AI techniques (Fu et al., 2022; Jiang et al., 2023; Wang et al., 2024a; Gan et al., 2023) from different perspectives to aid in the decipherment of Oracle Bone Script.

**Graph Retrieval-Augmented Generation for VLMs.** Although large-scale VLMs demonstrate strong zero-shot generalization, they still exhibit noticeable performance drops when the underlying training corpora lack or misrepresent the necessary domain knowledge (Zhang et al., 2024; Minaee et al., 2024). To enhance the specialization of visual language approaches in particular domains, Retrieval-Augmented Generation (RAG) approaches are employed (Lin, 2024; Zhang et al.,

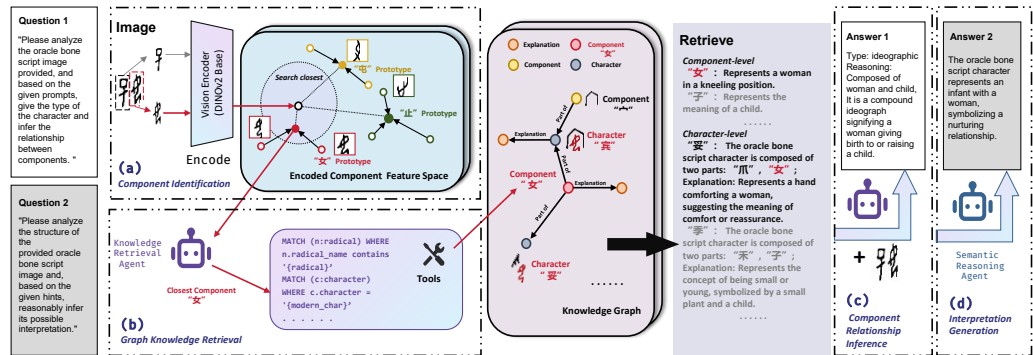

Figure 3: Detailed pipeline of our approach: (a) Component Identification Module identifies radical components from input OBS images; (b) Agent-Driven Graph Knowledge Retrieval retrieves relevant information from our constructed knowledge graph; (c) Component Relationship Inference uses VLMs to determine the structural relationships among components; (d) Interpretation Generation produces comprehensive semantic interpretations of oracle characters.

2025). Unlike traditional fine-tuning, RAG dynamically retrieves relevant knowledge from external databases during inference, enabling VLMs to access domain-specific information on-demand without updating their pre-trained parameters. Additionally, to mitigate the potential noise present in general knowledge bases that may affect results, the concise representation provided by knowledge graphs are integrated, forming what is known as Graph RAG (Peng et al., 2024).

**Character-level Oracle Dataset.** Existing datasets are designed primarily at the character level, focusing on complete characters. HUST-OBC (Wang et al., 2024b) and EVOBC (Guan et al., 2024a), as well as other representative datasets such as OBC306 (Huang et al., 2019), Oracle-50k (Han et al., 2020), and HWOBC (Li et al., 2020), provide large-scale collections of oracle bone characters and cover a wide range of historical periods including Oracle Bone Script, Warring States script, Seal Script, and Clerical Script. However, these datasets consist solely of full-character images without component-level annotations, limiting their utility for structural decomposition or semantic analysis aimed at understanding the internal organization of individual characters.

## 3 METHOD

As shown in Figure 3, our approach integrates visual analysis of OBS with structured knowledge reasoning through an agent-driven retrieval-augmented generation pipeline, comprising four parts: (1) a component identification module through character radicals retrieval as shown in Figure 3a, (2) an agent-driven knowledge graph retrieval module to dynamically query relevant entries as shown in Figure 3b, (3) a component relationship analysis and judgment module as shown in Figure 3c and (4) an interpretation generation module that integrates full character-level explanations as shown in Figure 3d. And the interpretation module supports two inference strategies: a VLM-based mode that directly fuses visual features with retrieved knowledge, and a multi-agent mode that separates retrieval and reasoning into specialized agents, enhancing robustness and interpretability.

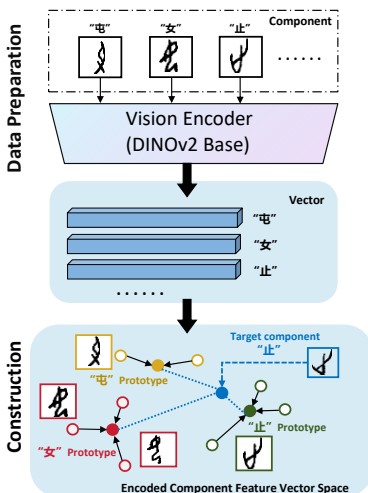

Figure 4: Construction of Vector Space.

### 3.1 COMPONENT IDENTIFICATION

To identify radical components from input OBS images, we first utilize a Vision Transformer (ViT) architecture based on DINOv2 (Dosovitskiy et al., 2021; Oquab et al., 2024) to construct a component feature space, as it produces highly transferable features. Then, we adopt a prototype-based

classifier following Prototypical Networks (Snell et al., 2017), as its class-level aggregation is well-suited to our low-data regime, improving robustness and reducing overfitting.

As shown in Figure 4, the input radical image $x$ is encoded by the DINOv2 encoder $f(\cdot)$ into a 768-dimensional vector $z$, and we then compute its prototype $p_c$ as the mean embedding of its support set $\mathbb{S}_c$ for each class $c$. Thus, given a query image $x_q$, its embedding $z_q = f(x_q)$ is compared to all class prototypes using Euclidean distance $d(\cdot, \cdot)$, and then classified into the class with the nearest prototype.

$$z = f(x), z \in \mathbb{R}^{768}; \quad \hat{y} = \arg\min_c \; d(z_q, p_c) \tag{1}$$

Compared with directly using conventional classifiers or detectors, this design enables our model to make efficient use of limited labeled samples and enhances generalization in the low-resource setting of OBS component identification.

## 3.2 Agent-Orchestrated Graph Knowledge Retrieval

We construct a Knowledge Graph (KG) from OB-Radix and character–component relations. For each test character, the PrototypeClassifier first predicts its most likely components; these predicted components are then used as *primary semantic cues* to query the KG. Rather than learning an unconstrained policy, we adopt a **cascading but largely fixed** retrieval pipeline, orchestrated by a tool-using LLM agent (Yao et al., 2023; Schick et al., 2023). The agent can call two external tools—*component explanation* and *characters-by-component*—and performs additional reasoning internally. Concretely:

- *Component-centric retrieval.* Given the predicted components, the agent first queries their explanations and searches for characters that contain these components, which typically provide the most direct semantic evidence.
- *Constrained synthesis.* When component-based retrieval yields weak or insufficient evidence, the agent internally performs variant lookup and modern–oracle mapping—without invoking external tools—to supplement the retrieved information. It then summarizes and reorders all evidence, both tool-obtained and internally inferred, into a concise, character-centric evidence bundle as input to the interpretation module.

To improve efficiency, we integrate a simple semantic-similarity cache following Jin et al. (2024), so that repeated or near-duplicate KG queries are served from cache. Overall, the agent acts as a lightweight orchestration layer over a deterministic retrieval cascade, ensuring that knowledge access is predictable and efficient while still providing rich, component-grounded context for downstream interpretation.

## 3.3 Component Relationship Inference

To move beyond black-box recognition, we design a module that leverages VLMs to infer the structural relationships among components. After the components are identified and the knowledge graph retrieval refines them, the system uses a VLM to jointly consider both visual embeddings and retrieved semantic information. The task requires the model to predict the inscription type of each oracle character, which can be categorized as ideographic, pictographic, or phono-semantic, and to generate a reasoning trace that explains how the components interact to form meaning. This process is illustrated in Figure 3, while the resulting output are presented in Figure 3c.

By conditioning the VLM on both structural and semantic cues, the module produces explanations that are not only accurate but also interpretable to human users. The component-level information is integrated into reasoning about character structure and provides the intermediate reasoning layer that connects recognition and interpretation generation.

## 3.4 Interpretation Generation

To generate full semantic interpretations of oracle characters, we design an inference pipeline that integrates visual recognition with knowledge-graph-based reasoning. Our framework supports two complementary modes of inference.

The first mode, *VLM Inference*, employs a VLM that jointly conditions on the visual embeddings of the inscription, component predictions from the PrototypeClassifier, and semantic prompts retrieved from the knowledge graph. By grounding ambiguous visual forms in curated historical evidence, the VLM produces interpretations that are semantically coherent and visually faithful.

Building upon this design, we further introduce a second mode, *Multi-Agent Inference*, inspired by recent advances in cooperative agent systems (Wu et al., 2023; Chang et al., 2024; Jin et al., 2025; Nguyen et al., 2025; Singh et al., 2025b; Wu et al., 2025). We use multi-agents to decouple retrieval and reasoning functions. A *Knowledge Retrieval Agent* plans and executes graph queries to gather relevant evidence, while a *Semantic Reasoning Agent* synthesizes this evidence with visual cues into structured, human-interpretable explanations. This separation improves robustness, reduces error propagation, and leverages the natural ability of large models to think after retrieval.

# 4 OUR COMPONENT-LEVEL ORACLE DATASET: OB-RADIX

Existing OBS datasets lack component-level annotations and expert verification, and contain a large number of misinterpretations. Therefore, working with paleographic experts, we collected more than 5,000 images of different OBSs from scratch based on high-quality transcriptions, ensuring clean and consistent visual inputs. After careful selection and organization by multiple paleographic Ph.D. students, we compiled a dataset containing 934 unique Oracle characters and 478 distinct components. In total, the dataset includes 1,022 character images and 1,853 component images, together with their corresponding semantic explanations. As illustrated in Figure 5, we manually segmented and annotated the component-level elements, including images and explanations.

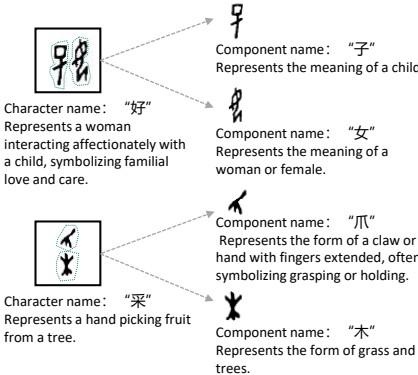

Figure 5: Our annotation of an oracle character at the component level.

# 5 EXPERIMENTS

To systematically evaluate that our approach achieves expert-level capability in OBS interpretation, we design a series of experiments under expert guidance, structured around three progressively advanced tasks: (1) component-level retrieval as the foundation, (2) component relationship inference as the intermediate stage, and (3) OBS interpretation generation as the ultimate goal.

## 5.1 METRICS AND BASELINES

We report ACC@k ($k \in \{1, 3, 5\}$) for component retrieval, and the accuracy of the oracle-character type classification for the component relationship inference experiment. And we employ BERTScore-F1, MoverScore, and OBS ROUGE-1 in OBS interpretation (Zhang* et al., 2020; Zhao et al., 2019; Altemeyer et al., 2025).

**OBS ROUGE-1.** However, relying solely on embedding-based semantic metrics (BERTScore-F1 and MoverScore) entails notable limitations: while they effectively capture the degree of semantic alignment between generated and reference texts, they often neglect lexical fidelity–that is, whether domain-critical terms or specific lexical items are accurately reproduced. Even within the

Table 1: Predefined Weights for Different POS Categories in Weighted ROUGE-1 Recall. Tags follow the Penn Treebank POS tagging standard.

| POS Category | Example Tags | Weight |
|---|---|---|
| Nouns | NN, NNS, NNP, NNPS | 1.0 |
| Verbs | VB, VBD, VBG, VBN, VBP, VBZ | 0.9 |
| Adjectives | JJ, JJR, JJS | 0.7 |
| Adverbs | RB, RBR, RBS | 0.7 |
| Function Words | PRP, PRP, DT, CC, IN | 0.3 |

highly specialized domain of OBS interpretation, standard ROUGE-1 proves inadequate: it assigns equal weight to all unigrams, failing to account for differences in their linguistic functions. This limitation is particularly critical, as accurate interpretation of oracle bone inscriptions typically depends on the precise usage of content words (e.g., nouns, verbs), rather than function words or grammatical particles.

Therefore, we introduce a weighted ROUGE metric, with coefficients calibrated through extensive consultations and iterative discussions with paleographers, to ensure that the scoring reflects the true semantic importance of different word classes in oracle bone interpretation. We use the NLTK toolkit (Bird et al., 2009) to tokenize and POS-tag both the reference and hypothesis, and assign predefined weights to their tags according to their importance in interpretation, as shown in Table 1. Formally, given a reference sentence $R$ and a hypothesis $H$, the OBS ROUGE-1 recall is defined as:

$$\text{OBS ROUGE-1}(R, H) = \frac{\sum_{\omega \in R \cap H} w(\text{POS}_\omega)}{\sum_{\omega \in R} w(\text{POS}_\omega)}. \tag{2}$$

Here $w(\text{POS}_\omega)$ denotes the weight assigned to the part-of-speech tag of word $\omega$.

In the experimental tables, we use shorthand notations for VLMs. Specifically, *GPT* refers to `GPT-5` (Singh et al., 2025a); *Claude* refers to `Claude Opus 4.1 (20250805)` (Anthropic, 2025); *GLM* refers to `GLM-4.5V` (V Team et al., 2025); and *Qwen* refers to `Qwen3-VL-235B-A22B` (Qwen Team, 2025).

## 5.2 DATASET SPLITTING

We adopted consistent dataset splitting strategies to ensure fair and realistic evaluation for all experiments. Specifically:

- **Component retrieval** (Section 5.3): Our OB-Radix dataset, containing 478 distinct components, was divided into training and testing sets with a ratio of 7:3, respectively. Model performance was measured by Top-1, Top-3, and Top-5 accuracy.
- **Component relationship inference** (Section 5.4): We constructed a seen set of 528 annotated instances, each including both inscription type labels and expert-derived reasoning traces. Models were trained and evaluated on this split without data overlap, ensuring interpretability analysis was grounded in expert references.
- **Interpretation generation** (Section 5.5): To avoid leakage, our KG was built using 70% of the corpus, while the remaining 30% was held out for testing. This split applies to all experiments related to Section 5.5. It guarantees that characters used for evaluation had not appeared in training, thus presenting a realistic challenge of interpreting previously unseen instances.

## 5.3 COMPONENT IDENTIFICATION

The most essential prerequisite for understanding oracle bone characters lies in the ability to accurately recognize their constituent components, since these components serve as the fundamental units from which higher-level semantic and structural interpretations are derived. As summarized in Table 2, our approach achieves competitive recognition accuracy, demonstrating its effectiveness in capturing the visual and structural properties of OBS. Representative recognition cases are illustrated in Figure 6.

Table 2: OBS component retrieval results.

| Metric | ACC↑ |
|--------|------|
| Top-1 | 0.7795 |
| Top-3 | 0.8855 |
| Top-5 | 0.9157 |

## 5.4 COMPONENT RELATIONSHIP INFERENCE

We evaluate whether VLMs capture the structural relationships among components, rather than treating OBS recognition as a black-box task. The task involves: (1) predicting the inscription type of a character (ideographic, pictographic, or phono-semantic), and (2) generating a textual explanation of component interactions. Representative examples comparing baseline and our enhanced pipeline are shown in Figure 7.

Table 3 reports classification and reasoning results. Our component-aware pipeline outperforms baselines across all metrics, confirming that explicit component-level knowledge improves both accuracy and interpretability. `Qwen3-VL-235B-A22B` achieves the highest classification accuracy (0.599), while `GPT-5` obtains the best reasoning similarity (BERTScore 0.670). `Claude Opus 4.1` further shows the strongest fluency and alignment in reasoning (MoverScore, OBS ROUGE-1).

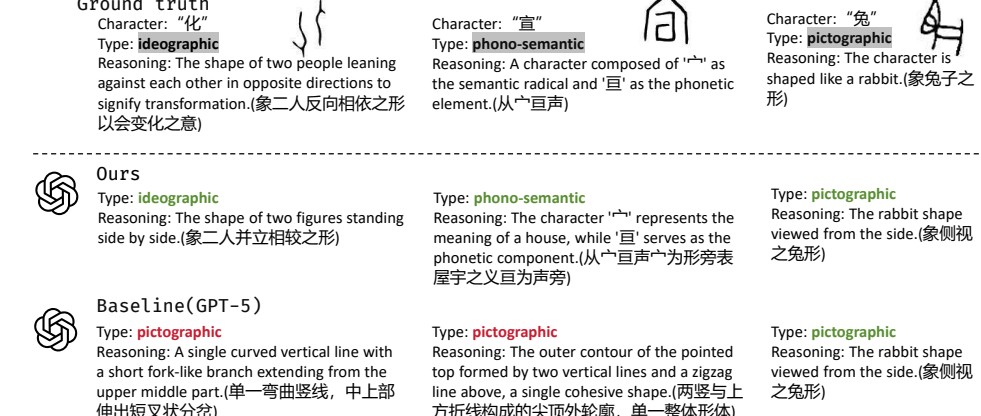

Figure 6: OBS Component identification examples.

Table 3: OBS component relationship inference results.

| Category | Model | ACC↑ | BERTScore↑ | MoverScore↑ | OBS ROUGE-1↑ |
|---|---|---|---|---|---|
| *Baseline* | GPT | 0.364 | 0.497 | 0.310 | 0.020 |
| | Claude | 0.475 | 0.495 | 0.324 | 0.016 |
| | GLM | 0.447 | 0.519 | 0.293 | 0.020 |
| | Qwen | 0.350 | 0.503 | 0.318 | 0.023 |
| *Ours* | GPT | 0.563 (+0.199) | **0.670** (+0.173) | 0.472 (+0.161) | 0.126 (+0.106) |
| | Claude | 0.551 (+0.075) | 0.648 (+0.152) | **0.490** (+0.166) | **0.160** (+0.144) |
| | GLM | 0.468 (+0.021) | 0.606 (+0.088) | 0.440 (+0.148) | 0.095 (+0.075) |
| | Qwen | **0.599** (+0.248) | 0.658 (+0.156) | 0.481 (+0.164) | 0.133 (+0.110) |

**Ground truth**

Character: "化"
Type: **ideographic**
Reasoning: The shape of two people leaning against each other in opposite directions to signify transformation.(象二人反向相依之形以会变化之意)

Character: "宣"
Type: **phono-semantic**
Reasoning: A character composed of '宀' as the semantic radical and '亘' as the phonetic element.(从宀亘声)

Character: "兔"
Type: **pictographic**
Reasoning: The character is shaped like a rabbit.(象兔子之形)

**Ours**

Type: **ideographic**
Reasoning: The shape of two figures standing side by side.(象二人并立相较之形)

Type: **phono-semantic**
Reasoning: The character '宀' represents the meaning of a house, while '亘' serves as the phonetic component.(从宀亘声宀为形旁表屋宇之义亘为声旁)

Type: **pictographic**
Reasoning: The rabbit shape viewed from the side.(象侧视之兔形)

**Baseline(GPT-5)**

Type: **pictographic**
Reasoning: A single curved vertical line with a short fork-like branch extending from the upper middle part.(单一弯曲竖线，中上部伸出短叉状分岔)

Type: **pictographic**
Reasoning: The outer contour of the pointed top formed by two vertical lines and a zigzag line above, a single cohesive shape.(两竖与上方折线构成的尖顶外轮廓，单一整体形体)

Type: **pictographic**
Reasoning: The rabbit shape viewed from the side.(象侧视之兔形)

Figure 7: Reasoning examples for component relationship inference. *Ground truth* shows expert interpretations; *Ours* and *Baseline* are model outputs, with correct answers in green and errors in red.

## 5.5 INTERPRETATION GENERATION

This task provides a direct test of whether the system can go beyond recognition and structural reasoning to generate semantically meaningful interpretations.

We compare two categories of approaches: (1) *Baseline* models, where LLMs directly generate interpretations without access to the Knowledge Graph; and (2) *Agentic RAG*, where the LLM retrieves supporting evidence from the graph before generating explanations. Performance was evaluated using OBS ROUGE-1, BERTScore, and MoverScore, with higher values indicating better alignment with expert-written ground truth. A concrete illustration is provided in A.3. Results are shown in Table 4.

The results clearly indicate the benefit of retrieval-augmented generation. Across all models, the Agentic RAG pipeline consistently outperforms the baseline counterparts. For example, `Qwen3-VL-235B-A22B` improves from 0.174 → 0.308 on OBS ROUGE-1 and from 0.362 → 0.471 on MoverScore. Similarly,

Table 4: OBS interpretation generation results.

| Category | Model | BERTScore↑ | MoverScore↑ | OBS ROUGE-1↑ |
|---|---|---|---|---|
| *Baseline* | GPT | 0.633 | 0.393 | 0.193 |
| | Claude | 0.614 | 0.365 | 0.197 |
| | GLM | 0.634 | 0.338 | 0.172 |
| | Qwen | 0.636 | 0.362 | 0.174 |
| *Agentic RAG (Ours)* | GPT | **0.727** (+0.094) | 0.475 (+0.082) | 0.317 (+0.124) |
| | Claude | 0.716 (+0.102) | 0.474 (+0.109) | **0.338** (+0.141) |
| | GLM | 0.706 (+0.072) | 0.453 (+0.115) | 0.265 (+0.093) |
| | Qwen | 0.722 (+0.086) | **0.471** (+0.109) | 0.308 (+0.134) |

`GPT-5` achieves the best BERTScore of 0.727 under the RAG setting, demonstrating stronger semantic alignment. These findings suggest that grounding interpretation generation in structured knowledge not only enhances factual accuracy but also produces outputs that are more coherent and interpretable.

## 5.6 ABLATION STUDY

To isolate the contribution of the Agent-Driven Graph Knowledge Retrieval, we conducted an ablation experiment in which retrieval was disabled and only component category predictions were provided. The results are summarized in Table 5.

Across all four models, the absence of retrieval consistently reduces performance, confirming that the Oracle Knowledge Graph supplies non-trivial

Table 5: OBS relationship inference results.

| Model | Retrieval Module | BERTScore↑ | MoverScore↑ | OBS ROUGE-1↑ |
|---|---|---|---|---|
| GPT | ✓ | **0.727** | **0.475** | 0.317 |
| | | 0.717(−0.010) | 0.469(−0.006) | 0.299(−0.018) |
| Claude | ✓ | 0.716 | 0.474 | **0.338** |
| | | 0.699(−0.017) | 0.451(−0.023) | 0.294(−0.044) |
| GLM | ✓ | 0.706 | 0.453 | 0.265 |
| | | 0.687(−0.019) | 0.415(−0.038) | 0.235(−0.030) |
| Qwen | ✓ | 0.722 | 0.471 | 0.308 |
| | | 0.711(−0.011) | 0.445(−0.026) | 0.271(−0.037) |

semantic context beyond visual recognition and component classification. Specifically, GPT-5 exhibits only a minor decline of approximately 1.1% on average, whereas the lighter GLM-4.5V suffers a drop of roughly 2.9%, indicating that retrieval acts as a knowledge-on-demand mechanism: larger models already internalize more domain knowledge, while smaller ones rely more heavily on external augmentation. Furthermore, the observed degradations span a bounded yet predictable range, from 0.6% (GPT-5 on MoverScore) to 4.4% (Claude Opus 4.1 on OBS ROUGE-1); this interval can serve as a diagnostic for future systems—ablation drops near the lower bound suggest the knowledge graph is approaching saturation, whereas values closer to the upper bound imply that refining retrieval content or query strategies may still yield additional gains.

## 5.7 MULTI-AGENT COLLABORATION

We further investigate a multi-agent setup, where the *Knowledge Retrieval Agent* first queries relevant entries from the Knowledge Graph, and the separate *Semantic Reasoning Agent* subsequently composes the interpretation (Figure 3d). This separation is motivated by our earlier findings that factual grounding

Table 6: Performance comparison of multi-agent configuration.

| Retrieval Agent | Reasoning Agent | BERT↑ | Mover↑ | OBS ROUGE-1↑ |
|---|---|---|---|---|
| Qwen3-VL-235B-A22B | DeepSeek-R1 | **0.760** | **0.507** | **0.265** |
| | GPT-5 | 0.705 | 0.445 | 0.231 |
| | Qwen3-235B-A22B | 0.734 | 0.470 | 0.225 |
| GPT-5 | DeepSeek-R1 | 0.733 | 0.476 | 0.246 |
| | GPT-5 | 0.713 | 0.458 | 0.240 |
| | Qwen3-235B-A22B | 0.729 | 0.454 | 0.235 |

and reasoning fluency benefit from distinct model capabilities. As shown in Table 6, the multi-agent configurations generally outperform single-agent baselines across the evaluated metrics. We hypothesize that the Semantic Reasoning Agent is better equipped to process and integrate the textual information retrieved from the KG, leveraging its specialized capabilities for enhanced coherence and accuracy.

## 5.8 HUMAN EXPERTS ASSESSMENT STUDY

To complement the above quantitative metrics, we conducted a human expert evaluation with two Ph.D. students in archaeology, using the 5-point Likert scale provided in A.4. For fairness, 10% of the held-out test set was selected, and participants were asked to evaluate the quality of generated interpretations along three pipelines: (1) the *Baseline pipeline* (direct generation using `Qwen3-VL-235B-A22B`), (2) the *RAG pipeline* (retrieval-augmented generation with `Qwen3-VL-235B-A22B`, and (3) the *Multi-Agent pipeline* (`Qwen3-VL-235B-A22B` as the Retrieval Agent and `DeepSeek-R1-250528` (Guo et al., 2025) as the Reasoning Agent).

The inter-rater reliability across all annotations was assessed using ICC3 (0.71) and Krippendorff's Alpha (0.74), indicating substantial to excellent agreement among two PhD evaluators with expertise in archaeology. Average Likert scores (on a scale of 1-5) revealed a clear performance hierarchy: the Multi-Agent Pipeline scored highest at 3.433, followed by the KG-RAG pipeline at 2.133, and the baseline pipeline at 1.367. These human evaluation results align with automatic metrics, confirming the robustness of our findings. The Multi-Agent Pipeline's score of 3.433—where 1 denotes poor quality and 5 indicates excellent, expert-like interpretations—demonstrates near-expert proficiency in archaeological artifact interpretation. To demonstrate the effectiveness of our multi-agent collaborative approach for oracle interpretation, we also qualitatively compare our approach with baseline methods in Figure 8.

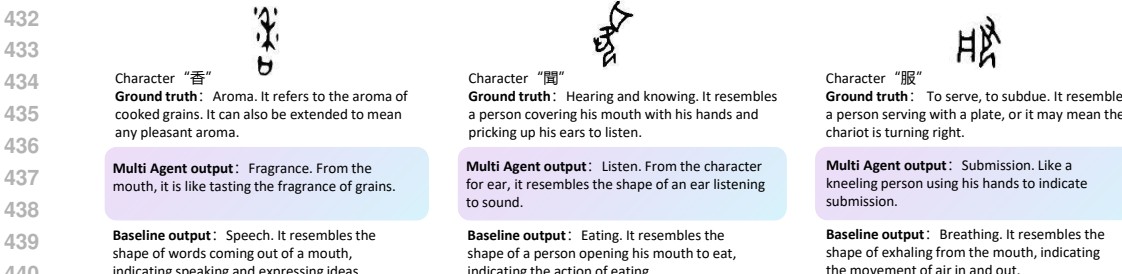

Figure 8: Comparison of approach outputs. *Character* displays the original Oracle bone characters; *Ground truth* provides the ground truth interpretations; *Multi Agent output* shows our multi agent approach's outputs using Graph RAG; *Baseline output* presents results from the baseline approach.

## 6 DISCUSSION

### 6.1 SUPPLEMENTARY EXPERIMENTS

In addition to the main experiments, we further conducted two supplementary studies to test the robustness and generalizability of our approach.

**English Interpretation Generation.** To investigate whether the models can generalize across languages, we constructed an English-version task, where the VLMs were required to output interpretations in English rather than Chinese. Results are reported in Table 7. Compared with the main Chinese results

Table 7: Results of interpretations conducted in English.

| Category | Model | BERTScore↑ | MoverScore↑ | OBS ROUGE-1↑ |
|---|---|---|---|---|
| **Baseline** | GPT-5 | 0.1517 | -0.1228 | 0.1099 |
| | Claude Opus 4.1 | 0.1356 | -0.1360 | 0.1192 |
| | GLM-4,5V | 0.0362 | -0.1732 | 0.0498 |
| | Qwen3-235B-A22B | 0.1587 | -0.1261 | 0.0843 |
| **Agentic RAG** | GPT-5 | 0.2717 | 0.0340 | 0.3220 |
| | Claude Opus 4.1 | 0.3177 | 0.0711 | 0.3797 |
| | GLM-4.5V | 0.3184 | 0.1062 | 0.2701 |
| | Qwen3-VL-235B-A22B | 0.3197 | 0.0753 | 0.2939 |

(Table 4), performance is notably lower across all metrics. This degradation is expected, since existing training corpora and retrieval databases are primarily constructed in Chinese, leading to weaker grounding in English. Nevertheless, the relative improvements of retrieval-augmented settings over baseline VLMs remain consistent, suggesting that our pipeline maintains cross-lingual robustness, albeit with a reduced ceiling. These results indicate the importance of developing parallel bilingual resources in paleographic studies to further support cross-linguistic generalization.

**Variant Character Recognition.** Oracle Bone Script often exhibits multiple variant forms for the same character, arising from alternative component structures or stylistic differences. To assess whether VLMs can identify such variants, we curated a set of 39 variant character pairs and prompted the models to

Table 8: Variant character search (39 samples)

| Model | Top-1@ACC | Top-5@ACC | Top-10@ACC |
|---|---|---|---|
| GPT-5 | 5.13% — 2 | 5.13% — 2 | 5.13% — 2 |
| Claude Opus 4.1 | 0.00% — 0 | 2.56% — 1 | 5.13% — 2 |
| GLM-4.5V | 2.56% — 1 | 2.56% — 1 | 2.56% — 1 |
| Qwen3-VL-235B-A22B | 2.56% — 1 | 2.56% — 1 | 5.13% — 2 |

determine which standard character each variant belonged to. As shown in Table 8, the overall accuracy remains low, with the best performance reaching only 5.13% Top-1 accuracy (2 correct out of 39). Even when relaxed to Top-10, none of the models exceeded 5.13%. These findings highlight that *variant recognition remains an open challenge*, likely due to the limited presence of variant forms in training data and the fine-grained visual distinctions required. Without targeted data augmentation and explicit modeling of variant–standard mappings, the models struggle to capture this essential dimension of paleographic reasoning.

Together, these supplementary experiments suggest that while retrieval-augmented models exhibit cross-lingual robustness, further efforts are needed to enhance sensitivity to historical character variants, which remain a critical bottleneck in Oracle Bone Script interpretation.

### 6.2 LIMITATIONS AND FUTURE DIRECTIONS

In collaboration with paleographic experts, we identify several limitations of the current pipeline. Component recognition is not always precise or complete, and the system may occasionally intro-

duce spurious elements. In addition, some oracle characters still lack widely accepted interpretations, which constrains the reliability of automated analysis.

Looking ahead, future work should focus on improving component recognition accuracy, enhancing the quality and coverage of the knowledge base, and extending the framework to better handle phono-semantic compounds. These directions would bring the system closer to expert reasoning practices and support more reliable AI-assisted paleography.

## 7 CONCLUSION

We propose a novel agent-driven framework that leverages the pictographic nature of OBS and the intrinsic relationships among components. Our approach integrates a component-structured Graph RAG with VLMs to advance OBS interpretation. We further construct a new component-level oracle dataset, enabling models to systematically capture the visual and structural properties of characters. In addition, we design three progressive tasks: component-level retrieval, component relationship inference, and script interpretation, which allow expert-level evaluation of model outputs and provide a more principled alternative to surface-level assessment. Experimental results demonstrate that coupling VLMs with knowledge graph augmentation through agent-based orchestration not only improves the accuracy and interpretability of OBS analysis but also underscores the potential value of this approach in other specific domains.

## 8 ETHICS STATEMENT

This work uses publicly available Oracle Bone Script (OBS) resources and contains no personal or sensitive data. All character- and component-level annotations were conducted with the assistance of paleographic experts and archaeology Ph.D. students to ensure accuracy. For human evaluation, two Ph.D. students participated voluntarily with informed consent. To reduce fatigue, only 10% of the held-out test set was assessed under consistent conditions using a standardized Likert scale.

We note that automatic interpretation of cultural heritage materials may introduce errors. Our dataset and results are intended solely as research aids to support, not replace, expert scholarship.

## 9 REPRODUCIBILITY STATEMENT

**Code and Data Availability**
All source code and the custom dataset (OB-Radix) are included in the submitted package. The dataset contains Chinese/English character explanations, component relationship annotations, and organized images. Detailed directory structure and usage instructions are provided in the README.

**Computational Requirements**
Experiments require Python 3.8+, PyTorch, Transformers, and other dependencies listed in `requirements.txt`.

**Reproducibility Instructions**
(1) Install dependencies with `pip install -r requirements.txt`. (2) Prepare the dataset using `python tools/sync_data.py`. (3) Run experiments from the corresponding subdirectories (see README for detailed commands). Evaluation scripts are provided for each experiment.

**Random Seeds and Determinism**
All experiments use fixed random seeds specified in configuration files, and GPU operations are set to be deterministic where supported.

**Baselines and Evaluation**
Baseline implementations share the same preprocessing and evaluation pipelines. Metrics are standardized across experiments for fair comparison.

**Limitations**
Some experiments may require significant computational resources or specific hardware. External APIs, if used, require appropriate keys.

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

## A APPENDIX

### A.1 USE OF LLM

We used large language models (LLM) to assist with language polishing and improving readability of portions of this manuscript, specifically OpenAI GPT-5 and xAI Grok 3. Additionally, we used Anthropic Claude Sonnent 3.7 to generate code snippets presented in this work. **Responsibility Statement**: The authors take full responsibility for all content of the manuscript, including portions edited and code generated by the LLMs. All generated code was reviewed and verified by the authors to ensure correctness and compliance with intended functionality. Any errors or misrepresentations in LLM-suggested text or code are the authors' responsibility.

### A.2 MORE DETAILS ON DATASET CONSTRUCTION

To ensure fine-grained component-level annotation, we adopted **LabelMe**[1] as the primary tool for manual segmentation of Oracle Bone Script images. LabelMe allows annotators to draw polygonal masks directly on images, making it well suited for the irregular shapes and complex outlines of Oracle characters, as shown in Figure 9.

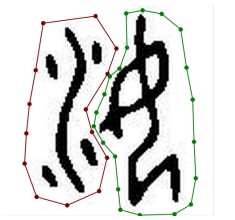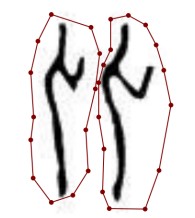

Figure 9: Two images of oracle bone characters segmented by labelme.

Each annotation task was conducted by archaeology PhD students who followed authoritative decipherment references. The process began with drawing precise polygons around each component within a character image. These polygons were then exported into JSON format, which stores the coordinates of the segmentation boundaries together with the corresponding component labels. To improve annotation consistency, we designed a standardized guideline specifying:

- **Segmentation granularity:** ensuring that even small components with distinct semantic functions were delineated separately.
- **Boundary precision:** refining polygon edges to closely follow character contours, especially in cases where strokes overlapped or eroded.
- **Label consistency:** using controlled vocabularies for component names to avoid ambiguity across annotators.

We further accelerated the process by employing large language models to provide preliminary segmentation suggestions, which annotators then carefully revised. As illustrated in Figure 9, the workflow produces both the original Oracle character and its corresponding component-level masks, which are subsequently paired with expert-verified semantic explanations.

This semi-automated, expert-curated procedure ensures that **OB-Radix** achieves both high annotation quality and interpretive reliability, laying the foundation for downstream tasks in component recognition and semantic inference.

### A.3 ORACLE BONE SCRIPT INTERPRETATION EXAMPLE

This section provides an example of oracle bone script (OBS) interpretation generated by our models to illustrate the difference between the *Baseline* and *Agentic RAG* approaches.

---

[1]https://github.com/wkentaro/labelme

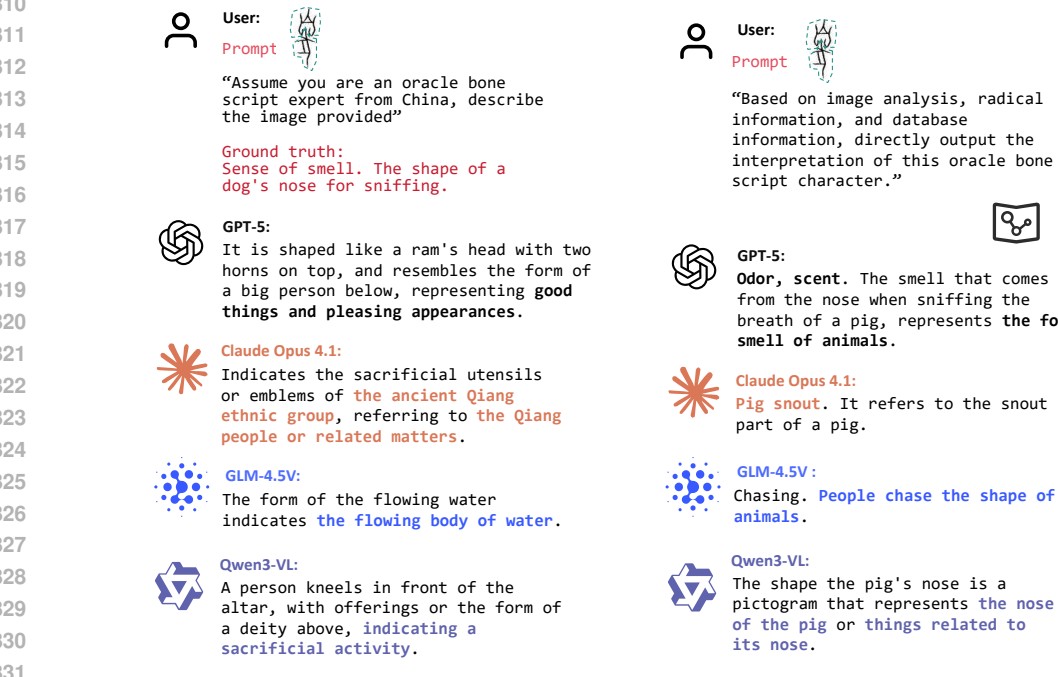

Figure 10: The left side are the baseline outputs, and the right side are ours.

## A.4 ORACLE BONE SCRIPT INTERPRETATION QUESTIONNAIRE

The questionnaire consists of 30 candidate interpretations of oracle bone script characters. Specifically, we curated 10 distinct characters, each of which is associated with three alternative interpretations reflecting different reasoning pipelines. To avoid introducing bias from fixed presentation sequences, the three interpretations corresponding to the same character were randomly permuted prior to distribution. This randomization was applied independently across pipelines, ensuring that participants evaluated the interpretations without being influenced by a consistent order effect. Consequently, the design of the questionnaire facilitates a more balanced and reliable assessment of the comparative quality of the proposed interpretation methods.

## Instructions

Do you agree with this interpretation of the oracle bone script? (5-point scale) Please tick (√) the score that best reflects your agreement with each oracle-bone-script interpretation below.

## Scoring Scale

• **(5) Completely Agree**: The interpretation fully matches the oracle bone script's glyph original meaning and construction logic, without any semantic distortion or historical deviation.

• **(4)Basically Agree**: The core interpretation is correct (matches the glyph original meaning and mainstream views), but there are extremely minor expression flaws (such as imprecise terminology) or omissions of secondary information (such as not mentioning rare usages), which do not affect the overall accuracy of the interpretation.

• **(3) Neutral**: The interpretation has "ambiguity" or "points of controversy" there is no clear evidence to prove it wrong, nor does it fully match authoritative interpretations; possibly due to the oracle bone script's own glyph defects, ongoing academic debates, or the interpretation only covering partial possibilities.

• **(2) Basically Disagree**: The core interpretation is wrong (violates the glyph original meaning or mainstream academic views), but there are a few reasonable elements (such as correct partial glyph disassembly, or involving secondary usages of the character); the overall interpretation deviates from the essence, but not completely baseless.

• **(1) Completely Disagree**: The interpretation completely contradicts the oracle bone script's glyph, construction logic, and academic consensus; unrelated to any known usage of the character.

## Example

| | Output | Score | | | | |
|---|---|---|---|---|---|---|
| | | 1 | 2 | 3 | 4 | 5 |
|  | • It is like placing something on a stand with two hands. Four hands hold the object and place it on the ground or on a stand, which means to place or put it. | ○ | ○ | ○ | ○ | ○ |
| | • The image of holding jade in both hands and offering it to the altar represents a sacrificial ceremony. | ○ | ○ | ○ | ○ | ○ |
| | • It resembles four hands holding up a tube. | ○ | ○ | ○ | ○ | ○ |

Figure 11: Questionnaire

