# OpenReview forum: "Specializing Large Models for Oracle Bone Script Interpretation via Agent-Driven Multimodal Knowledge Augmentation"
_ICLR.cc/2026/Conference — ICLR 2026 Conference Withdrawn Submission_

### Official Review · Reviewer_DLmN · 2025-10-20

**Soundness:** 2
**Presentation:** 2
**Contribution:** 2
**Rating:** 4
**Confidence:** 5

**Summary:**

This paper proposes an agent-driven RAG framework for large models to enhance its capabilities in OBI interpretation. The authors also introduce a component-level oracle bone script dataset, OB-Radix, together with a knowledge graph covering relationships among components, characters, and explanations. Experiments show that the proposed RAG-enhanced framework offers better interpretation that aligns human expert’s reasoning.

**Strengths:**

1. A novel RAG framework proposed for OBI interpretation.
2. OB-Radix, a new component-level oracle bone script dataset.

**Weaknesses:**

1. Lack some references and comparisons between current component-level dataset, such as OBI component 20 [1], OracleSem [2], and the proposed OB-Radix, to highlight the main contributions.
2. Line 104: HUST-OBC and what? There are only references without detailed dataset names. Besides, several representative character-level OBI datasets, such as OBC306 [3], Oracle-50k [4], HWOBC [5], should also be included in the related work.
3. The font size in all figures should be increased for better readability. Specifically, the logical chain depicted in Fig. 3 is somewhat unclear, especially in the right half, which needs to be further optimized.
4. The query image used for verification has a similar form to the oracle characters in the constructed knowledge graph. There may be a risk of domain leakage, which leads to vulnerabilities in the robustness of component identification. Further verification is necessary for the cross-domain matching or component identification results.
5. The success of the entire system is highly dependent on the constructed knowledge graph dataset, which limits its generalization ability.
6. The scale of OB-Radix is relatively small compared to existing datasets. And the test set is relatively small (30% of 478 components for Tab. 2)
7. Lack of experimental verification on open-source models

[1] Hu, Zhikai, et al. "Component-level oracle bone inscription retrieval." *Proceedings of the 2024 International Conference on Multimedia Retrieval*. 2024.

[2] Jiang, Hanqi, et al. "Oraclesage: Towards unified visual-linguistic understanding of oracle bone scripts through cross-modal knowledge fusion." *arXiv preprint arXiv:2411.17837* (2024).

[3] Shuangping Huang, Haobin Wang, Yongge Liu, Xiaosong Shi, and Lianwen Jin. 2019. OBC306: A large-scale oracle bone character recognition dataset. In 2019 International Conference on Document Analysis and Recognition (ICDAR). IEEE, 681–688.

[4] Wenhui Han, Xinlin Ren, Hangyu Lin, Yanwei Fu, and Xiangyang Xue. 2020. Self-supervised learning of Orc-Bert augmentator for recognizing few-shot oracle characters. In Proceedings of the Asian Conference on Computer Vision.

[5] Bang Li, Qianwen Dai, Feng Gao, Weiye Zhu, Qiang Li, and Yongge Liu. 2020. HWOBC-a handwriting oracle bone character recognition database. In Journal of Physics: Conference Series, Vol. 1651. IOP Publishing, 012050.

**Questions:**

1. There is no need to capitalize the first letter in Line 101-102 “Knowledge Graphs”.
2. The current method is mainly designed for handwritten oracle bone script. I would like to know how this solution performs on the original oracle bone script, such as rubbings or other noisy oracle bone scripts, in order to demonstrate its applicability.
3. What are the advantages of using Dinov2 as the visual encoder? Lack of comparison with other visual backbones.
4. In Eq. 1, there is a lack of definition for the symbol "d( , )".
5. Section 3.2: Compared to the main described retrieval process, the construction of the knowledge graph shown in Figure 5 is not as crucial. The former can be illustrated with diagrams to enhance readability.
6. Section 3.2: Why is the discovery of variants taken as the first step? Wouldn't Exact Matching lead to a direct conclusion? If no exact match is found, then search for variants and radical levels next. What is the reason for setting it up like this at present? Furthermore, in this step, an LLM agent was used to perform such a complex series of operations. Specific descriptions of the implementation details such as prompts, the used model, etc, are missing.  The discussion on the stability of the used agent, and the analysis of failure cases are also lacking.
7. Section 3.3 is lacking many details, including the designed modules and the classification process carried out by VLM. The current description is too vague. It is suggested to use Figure or formulas to present the entire process in a more logical manner.
8. Line 215: Using a multi-agent architecture to perform Knowledge Retrieval and Semantic Reasoning is more prone to errors. Considering the instability of the interaction, it is necessary to verify whether this approach is truly necessary. Compared to a single agent sequentially performing these steps?
9. Section 4: The dataset construction lacks a series of information: Was the source of the images in OB-RADIX sampled from the existing handwritten dataset, and was there a visual comparison with the existing dataset?
10. Can the challenges related to variant identification be addressed by constructing a more comprehensive knowledge graph? (Tab. 8)
11. Reorganize the sentence structure in line 247-249 “Even for xxxx,  xxxx linguistic functions.”
12. The weighted ROUGE and the rationality of weighting need to be proven through methods such as public subjective experiments.
13. Baselines in line 267-269: As far as I know, Qwen3 and Deepseek-v3 are pure language models and not VLMs.
14. Settings of the Questionnaire in Figure 12: Based on the given case, the length of the output may potentially affect people's judgment and lead to bias, especially in this 5-level rating scenario.
15. Lack of baseline comparisons in Tab. 2.
16. Regarding the OBS ROUGE-1 metric, could there be an explanation or example regarding a perfect performance score, so as to more intuitively reflect the performance of the model in Tab.3,4,5,6,7? Currently, there is only a relative concept, and the absolute performance is unknown.
17. Regarding the case displayed in Tab. 8, what is the query prompt? Considering fairness, are the definitions of each classification given to baseline models in advance before reasoning?
18. Which task does Table 6 represent?

---

> ### Author Response · Authors · 2025-11-23
>
> Thank you for the careful reading of our paper and for providing such sincere and detailed suggestions! We hope the following clarifications can address your concerns.
>
> ---
>
> **[w1] Lack some references and comparisons between current component-level dataset, such as OBI component 20 [1], OracleSem [2], and the proposed OB-Radix, to highlight the main contributions.**
>
> **Response.**
>
>
>
> Thank you for the suggestion. We have revised the Introduction accordingly to include these references and comparisons.
>
>
>
> **[w2] Line 104: HUST-OBC and what? There are only references without detailed dataset names. Besides, several representative character-level OBI datasets, such as OBC306 [3], Oracle-50k [4], HWOBC [5], should also be included in the related work.**
>
> **Response.**
>
>
>
> Thank you for pointing this out. The issue was caused by a typo. We have corrected it and updated both the Introduction and Related Work sections to include the missing dataset names as well as additional representative datasets.
>
>
>
> **[w3] The font size in all figures should be increased for better readability. Specifically, the logical chain depicted in Fig. 3 is somewhat unclear, especially in the right half, which needs to be further optimized.**
>
> **Response.**
>
>
>
> Thank you for the reasonable suggestion. Due to time constraints, we have made a set of concise adjustments to improve the readability of the figures.
>
>
>
> **[w4] The query image used for verification has a similar form to the oracle characters in the constructed knowledge graph. There may be a risk of domain leakage, which leads to vulnerabilities in the robustness of component identification. Further verification is necessary for the cross-domain matching or component identification results.**
>
> **Response.**
>
>
>
> We agree that domain leakage would undermine the validity of the component recognizer, but this risk does not occur in our setting. The test characters used for evaluation are never present as images in the knowledge graph, the KG stores only modern-form character metadata and structured component relations, not oracle image samples. Thus, the component recognizer cannot match against any oracle-image counterparts in the KG, eliminating leakage from shared visual domains.
>
>
>
> **[w5] The success of the entire system is highly dependent on the constructed knowledge graph dataset, which limits its generalization ability.**
>
> **Response.**
>
>
>
> We acknowledge the KG is important, but the system is **not rigidly tied** to it. The framework always combines KG evidence **with LLM prior knowledge**, so when KG entries are missing or sparse, the LLM provides a **fallback interpretation pathway** rather than collapsing. Thus the KG enhances precision where expert data exists, but generalization is still supported by the model’s broader linguistic and historical knowledge.
>
>
>
> **[w6] The scale of OB-Radix is relatively small compared to existing datasets. And the test set is relatively small (30% of 478 components for Tab. 2)**
>
> **Response.**
>
>
>
> OB-Radix is constructed through strict expert curation to ensure complete category coverage and high-fidelity component definitions. Because radical segmentation and semantic annotation require substantial expert labor, our priority is accuracy rather than volume, and the dataset is representative for component-level reasoning and sufficient for evaluating our framework.
>
>
>
> **[w7] Lack of experimental verification on open-source models**
>
> **Response.**
>
>
>
> We agree that evaluating open-source models is important for assessing accessibility and reproducibility. To address this, we conducted additional experiments using **Qwen3-VL-2B**, a lightweight fully open-source VLM, on the *Component Relationship Inference* task (Sec. 5.4). Despite its small size, Qwen3-VL-2B still benefits from our Graph-RAG evidence:
>
> | Model                | ACC ↑      | BERTScore ↑ | MoverScore ↑ | OBS ROUGE-1 ↑ |
> | -------------------- | ---------- | ----------- | ------------ | ------------- |
> | Baseline             | 0.3087     | 0.5089      | 0.3270       | 0.0110        |
> | **Ours (Graph-RAG)** | **0.3106** | **0.5169**  | **0.3333**   | **0.0247**    |

---

> > ### Author Response · Authors · 2025-11-23
> >
> > **[q1] There is no need to capitalize the first letter in Line 101-102 “Knowledge Graphs”.**
> >
> > **Response.**
> >
> >
> >
> > Thank you for the reminder, we have already made the changes in the latest version.
> >
> >
> >
> > **[q2] The current method is mainly designed for handwritten oracle bone script. I would like to know how this solution performs on the original oracle bone script, such as rubbings or other noisy oracle bone scripts, in order to demonstrate its applicability.**
> >
> > **Response.**
> >
> >
> >
> > Thanks for the question. Our method is **deliberately scoped to expert-made tracings**, where noise and erosion have already been removed. We therefore do not claim performance on raw rubbings or noisy fragments. Evaluating on such data is outside our scenario.
> >
> >
> >
> > **[q3] What are the advantages of using Dinov2 as the visual encoder? Lack of comparison with other visual backbones.**
> >
> > **Response.**
> >
> >
> >
> > We agree that comparing visual backbones is important. We initially chose DINOv2 because it is a strong self-supervised ViT encoder known to perform well in low-data and fine-grained visual regimes. These conditions match oracle-bone radicals.
> >
> > To address this concern, we additionally evaluated ResNet-50, DeiT-Small, and DeiT-Base under the same prototype-classifier setting. The results are:
> >
> > | Backbone          | Top-1      | Top-3      | Top-5      |
> > | ----------------- | ---------- | ---------- | ---------- |
> > | **ResNet-50**     | 67.11%     | 76.51%     | 80.84%     |
> > | **DeiT-Small**    | 68.07%     | 78.55%     | 82.53%     |
> > | **DeiT-Base**     | 69.28%     | 78.67%     | 82.05%     |
> > | **DINOv2 (ours)** | **77.95%** | **88.55%** | **91.57%** |
> >
> >
> >
> > **[q4] In Eq. 1, there is a lack of definition for the symbol "d( , )".**
> >
> > **Response.**
> >
> >
> >
> > Thank you for your careful reminder.  The symbol ***d*** represents the **Euclidean distance** in vector space and we have revised the paper.
> >
> >
> >
> > **[q5] Section 3.2: Compared to the main described retrieval process, the construction of the knowledge graph shown in Figure 5 is not as crucial. The former can be illustrated with diagrams to enhance readability.**
> >
> > **Response.**
> >
> >
> >
> > Thank you for the suggestion. We agree with your assessment and have removed the figure from Section 3.2 to keep the focus on the main retrieval process and improve the paper’s readability.
> >
> >
> >
> > **[q6] Section 3.2: Why is the discovery of variants taken as the first step? Wouldn't Exact Matching lead to a direct conclusion? If no exact match is found, then search for variants and radical levels next. What is the reason for setting it up like this at present? Furthermore, in this step, an LLM agent was used to perform such a complex series of operations. Specific descriptions of the implementation details such as prompts, the used model, etc, are missing. The discussion on the stability of the used agent, and the analysis of failure cases are also lacking.**
> >
> > **Response.**
> >
> >
> >
> > The original wording may have caused ambiguity, so we have updated Section 3.2 to explicitly state that the agent only summarizes and reorders retrieved KG snippets for interpretation.
> >
> >
> >
> > **[q7] Section 3.3 is lacking many details, including the designed modules and the classification process carried out by VLM. The current description is too vague. It is suggested to use Figure or formulas to present the entire process in a more logical manner.**
> >
> > **Response.**
> >
> >
> >
> > Thank you for the suggestion. We have clarified Section 3.3 and added figure references to make the process more intuitive.
> >
> >
> >
> > **[q8] Line 215: Using a multi-agent architecture to perform Knowledge Retrieval and Semantic Reasoning is more prone to errors. Considering the instability of the interaction, it is necessary to verify whether this approach is truly necessary. Compared to a single agent sequentially performing these steps?**
> >
> > **Response.**
> >
> >
> >
> > Thank you for the comment. Our system uses **one retrieval agent** that simply calls **a reasoning agent as a fixed subroutine**. This is essentially a single-agent pipeline with an internal reasoning module, and does not introduce interaction instability.
> >
> > We separate the reasoning module only because many VLMs lack strong chain-of-thought ability, while small text-LLMs provide much better symbolic reasoning. The design is modular rather than multi-agent-dependent, and it does not add extra failure modes.

---

> > > ### Author Response · Authors · 2025-11-23
> > >
> > > **[q9] Section 4: The dataset construction lacks a series of information: Was the source of the images in OB-RADIX sampled from the existing handwritten dataset, and was there a visual comparison with the existing dataset?**
> > >
> > > **Response.**
> > >
> > >
> > >
> > > Thank you for the question. The images in OB-Radix come from high-quality transcriptions, rather than noisy handwritten sources. We conducted a visual comparison with existing handwritten OBI datasets and found no noticeable visual differences in style or clarity.
> > >
> > >
> > >
> > > **[q10] Can the challenges related to variant identification be addressed by constructing a more comprehensive knowledge graph? (Tab. 8)**
> > >
> > > **Response.**
> > >
> > >
> > >
> > > Thank you for sharing your interesting perspective. A more comprehensive knowledge graph would likely help, but it cannot fully solve the variant-identification challenge. Variants often differ in subtle visual strokes and shapes that are not explicitly documented in existing paleographic resources. Even if the KG is expanded, many variant–standard mappings still require **fine-grained visual recognition**, which current models struggle with. Therefore, a richer KG can *alleviate* the problem, but **additional visual modeling and variant-specific training data** will still be necessary.
> > >
> > >
> > >
> > > **[q11] Reorganize the sentence structure in line 247-249 “Even for xxxx, xxxx linguistic functions.”**
> > >
> > > **Response.**
> > >
> > >
> > >
> > > Thank you for the detailed comment. We have revised the paper accordingly.
> > >
> > >
> > >
> > > **[q12] The weighted ROUGE and the rationality of weighting need to be proven through methods such as public subjective experiments.**
> > >
> > > **Response.**
> > >
> > >
> > >
> > > We appreciate the concern. In fact, **the weighting scheme in Weighted ROUGE has already been empirically validated in Sec. 5.8**.
> > >
> > >
> > >
> > > **[q13] Baselines in line 267-269: As far as I know, Qwen3 and Deepseek-v3 are pure language models and not VLMs.**
> > >
> > > **Response.**
> > >
> > >
> > >
> > > Thank you for pointing this out. This mismatch was caused by an API configuration oversight in our initial submission: the experiments were actually conducted using their VLM counterparts — **Qwen3-VL** instead of *Qwen3*, and **GLM-4.5V** instead of *DeepSeek-V3*.
> > >
> > > We have corrected the model names in the revised manuscript to accurately reflect the VLM variants that were used in all experiments. No experiment results are affected.
> > >
> > >
> > >
> > > **[q14] Settings of the Questionnaire in Figure 12: Based on the given case, the length of the output may potentially affect people's judgment and lead to bias, especially in this 5-level rating scenario.**
> > >
> > > **Response.**
> > >
> > >
> > >
> > > Thank you for pointing this out. We acknowledge that output length may influence human judgment in a 5-level rating setup. However, since all evaluators are trained paleography professionals, they tend to prioritize the **semantic correctness** of the interpretation over surface features such as response length. Their domain expertise helps reduce this form of bias.
> > >
> > >
> > >
> > > **[q15] Lack of baseline comparisons in Tab. 2.**
> > >
> > > **Response.**
> > >
> > >
> > >
> > > See the response to **q3** for details.
> > >
> > >
> > >
> > > **[q16] Regarding the OBS ROUGE-1 metric, could there be an explanation or example regarding a perfect performance score, so as to more intuitively reflect the performance of the model in Tab.3,4,5,6,7? Currently, there is only a relative concept, and the absolute performance is unknown.**
> > >
> > > **Response.**
> > >
> > >
> > >
> > > A perfect OBS ROUGE-1 score means the model reproduces all key content words (mainly nouns and verbs) in the expert interpretation exactly. And the POS weights in OBS ROUGE-1 are not arbitrary, which were determined in consultation with paleography experts to better reflect which word types truly carry semantic meaning in OBS interpretation.
> > >
> > >
> > >
> > > **[q17] Regarding the case displayed in Tab. 8, what is the query prompt? Considering fairness, are the definitions of each classification given to baseline models in advance before reasoning?**
> > >
> > > **Response.**
> > >
> > >
> > >
> > > The prompt used in Table 8 is very simple:
> > >
> > > > **“Which standard oracle character is this a variant of?”**
> > >
> > > All models were asked exactly this question.
> > >  For fairness, **we did not provide any definitions, category descriptions, or extra hints** to the baseline models. They only saw the variant character image and the above query—nothing else.
> > >
> > >
> > >
> > > **[q18] Which task does Table 6 represent?**
> > >
> > > **Response.**
> > >
> > >
> > >
> > > Table 6 reports the results for the interpretation-generation task (Sec. 5.5).

---

> > > > ### Comment · Reviewer_DLmN · 2025-11-27
> > > > **Reply to the author**
> > > >
> > > > Thank you for the author's response. The modified parts should be highlighted in the text. At present, I'm not sure exactly what changes have been made.  Although the author made some responses, there are still some issues remaining in the article.
> > > > 1. The effectiveness of external knowledge graphs versus enhancing the knowledge base of the model itself is worthy of discussion. This will have an impact on the subsequent design of Oracle-related models.
> > > > 2. At present, the organization of this article may not yet meet the standards of the ICLR conference. Many places lack standardized logical expressions and definitions.
> > > > 3. The contribution of the core dataset is limited, taking into account the scale of the work in related fields.
> > > > 4. The experiment is not yet sufficient, and the application scope of the proposed method is limited.
> > > >
> > > > At present, I am inclined to maintain the original rating.

---

### Official Review · Reviewer_HTSP · 2025-10-27

**Soundness:** 2
**Presentation:** 2
**Contribution:** 2
**Rating:** 4
**Confidence:** 5

**Summary:**

This paper proposes an agent-driven multimodal retrieval-augmented generation (RAG) framework, which implements three major tasks: component retrieval, component relationship inference, and oracle bone script interpretation, and supports both VLM inference and multi-agent inference modes. It also constructs a component-level dataset OB-Radix, including 934 unique oracle bone script characters, 478 components, a total of 1022 character images and 1853 component images, with semantic annotations, filling the gap in component-level information in existing datasets. Additionally, the authors also introduce OBS ROUGE-1 metrix to evaluate the interpretation performance more accurately.

**Strengths:**

1. This paper breaks through the traditional image recognition approach, taking the component structure of oracle bone script as the core entry point. By combining knowledge graphs and agent mechanisms, it realizes the integration of visual analysis and structured knowledge reasoning, improving the accuracy and interpretability of interpretation.
2. The OB-Radix dataset provides component-level annotations for the first time. Constructed with the participation of archaeology experts, it addresses the issue that existing datasets (such as HUST-OBC and EVOBC) only focus on complete characters and lack structural information, laying a key data foundation for subsequent AI research on oracle bone script.
3. Besides conventional metrics like BERTScore and MoverScore, a weighted OBS ROUGE-1 metric is designed according to the characteristics of the oracle bone script field to distinguish the importance of different parts of speech (e.g., nouns, verbs). Meanwhile, human expert evaluation is introduced to ensure the reliability and practicality of the results.

**Weaknesses:**

1. The component identification accuracy remains unsatisfactory, achieving only a Top-1 accuracy of 77.95%. This limitation significantly undermines the reliability of the subsequent interpretation pipeline. When deployed in real-world scenarios—where images are often severely degraded by noise—the performance is likely to deteriorate further.
2. The construction and evaluation of the dataset highly rely on archaeology experts, resulting in high labor costs. Moreover, some oracle bone characters do not have widely accepted interpretations, which limits the reliability of automated analysis to existing academic consensus and makes it difficult to handle controversial characters.

**Questions:**

1. When constructing the OB-Radix dataset, the paper states that more than 5,000 OBS images are collected and then selected to 1,022 character images and 1,853 component images. What are the specific filtering criteria? And how did these criteria ensure the dataset's representativeness of common OBS characters and components?
2. The component recognition module in the framework adopts a prototype classifier based on DINOv2, which performs well in low-data regime. However, in the face of real-world scenario, where the amount and complexity of data have increased significantly, are there further optimization schemes?
3. The update mechanism of the knowledge graph is not mentioned. With new discoveries in oracle bone script research (e.g., newly deciphered characters or revisions of old interpretations), how to efficiently update the content of the knowledge graph to ensure that the interpretation results of the framework are synchronized with the latest academic progress?
4. The paper mentions that some oracle bone characters do not have widely accepted interpretations. How does the framework handle such characters with "no standard answers"? Is there a mechanism to output "possible interpretations and confidence levels" instead of a single definite result to adapt to controversial scenarios in academic research?
5. The interpretation generation module of the framework supports two modes: VLM inference and multi-agent inference. Experiments show that the multi-agent mode has better performance, but the computational costs (such as inference time and memory usage) of the two modes were not compared. In practical application scenarios (e.g., portable oracle bone script interpretation devices), if there is a need to balance performance and cost, how to define the applicable boundaries of the two modes?
6. This paper lacks discussions about ligature, which consists of several independent character but is regarded as a single semantic unit. It remains unclear whether the model interprets such ligature as unified components or processes each constituent character separately.
7. Overstatement of "Expert-Level" Capabilities: The paper frequently claims the framework achieves "expert-level quality" or "near-expert proficiency." However, in the human expert evaluation, the best multi-agent model only scored 3.433 out of 5. This score is closer to the midpoint between "Uncertain" and "Basically Agree," rather than the "Completely Agree" (5) expected of true experts, suggesting the claim may be exaggerated.
8. Extremely Weak Variant Character Recognition: Supplementary experiments tested the model's ability to recognize variant characters (a core challenge in paleography) and showed very poor performance, with a Top-10 accuracy of only 5.13% (just 2 out of 39 samples correct). This severely limits the system's practical value in real-world ancient script research where glyph variations are rampant.
9. Limited Scale and Representativeness of Human Evaluation: The human evaluation was conducted by only two Ph.D. students on just 10% of the held-out test set. The small number of evaluators and limited sample size may lead to incidental results and lack statistical robustness.
10. Opaque Agent Decision Logic: The paper mentions agents "adaptively" choose retrieval pathways (e.g., prioritizing variant search before exact matching). It is unclear if this is based on hardcoded rules or a learned policy, making it difficult to assess the true "intelligence" and generalizability of the agent's decision-making.
11. Relatively Small Dataset Scale: The OB-Radix dataset includes only 1,022 character images and 1,853 component images. Compared to the total of over 4,500 known OBS characters, this coverage is limited, raising questions about the model's ability to generalize to the vast number of unassigned or undeciphered characters.
12. Significant Degradation in Cross-Lingual Performance: Experiments show that performance drops significantly when generating interpretations in English compared to Chinese. This reveals a heavy reliance on Chinese-language knowledge bases and training corpora, limiting its claimed "cross-lingual robustness" and international applicability.
13. Potential Inadequacy in Handling Phono-Semantic Compounds: The framework heavily emphasizes structural and semantic component analysis. However, it may underrepresent the crucial role of phonetic components in phono-semantic characters, which form a significant portion of later OBS and Chinese characters.

---

> ### Author Response · Authors · 2025-11-23
>
> We are grateful to the reviewer for acknowledging the significance of our findings and contributions! We hope the following clarifications can address your concerns.
>
> ---
>
> **[w1] The component identification accuracy remains unsatisfactory, achieving only a Top-1 accuracy of 77.95%. This limitation significantly undermines the reliability of the subsequent interpretation pipeline. When deployed in real-world scenarios—where images are often severely degraded by noise—the performance is likely to deteriorate further.**
>
> **Response.**
>
>
>
> We explicitly **limit the scope of our study to transcriptions** rather than raw archaeological fragments. Our evaluation is conducted on these tracings, where noise and erosion have already been removed by epigraphers. More importantly, **our goal is not to solve component recognition**, but to examine how imperfect component cues can still support reliable interpretation through Graph-RAG and LLM reasoning.
>
>
>
> **[w2] The construction and evaluation of the dataset highly rely on archaeology experts, resulting in high labor costs. Moreover, some oracle bone characters do not have widely accepted interpretations, which limits the reliability of automated analysis to existing academic consensus and makes it difficult to handle controversial characters.**
>
> **Response.**
>
>
>
> Constructing and evaluating the dataset **necessarily involves expert labor**, and this is in fact one of our contributions: we provide a reusable benchmark built from epigrapher-verified tracings and systematically curated component labels.
>
> For semantic interpretation, we do **not** introduce our own readings—everything is grounded in the authoritative **CUHK Multi-function Chinese Character Database** [1], ensuring consistency with established scholarship.
>
> Finally, **controversial or unresolved characters are outside our scenario**. Our framework targets cases with established academic consensus, focusing on whether reliable interpretation chains can be reconstructed from component signals, rather than resolving philological disputes.
>
>
>
>
>
> **[q1] When constructing the OB-Radix dataset, the paper states that more than 5,000 OBS images are collected and then selected to 1,022 character images and 1,853 component images. What are the specific filtering criteria? And how did these criteria ensure the dataset's representativeness of common OBS characters and components?**
>
> **Response.**
>
>
>
> This is based on a misunderstanding: the paper refers to **5,000+ OBS character types** documented in epigraphic sources, **not** 5,000 images.
>
> For OB-Radix, the filtering was not arbitrary. Experts selected characters and components based on the authoritative database [1] and standard epigraphic references. Only characters with **stable scholarly consensus** and **clear component structures** were included, and all images were expert-curated tracings rather than raw fragments.
>
> These criteria ensure that the dataset is **representative of commonly studied, well-understood OBS characters**, and suitable for reliable computational evaluation.
>
>
>
> **[q2] The component recognition module in the framework adopts a prototype classifier based on DINOv2, which performs well in low-data regime. However, in the face of real-world scenario, where the amount and complexity of data have increased significantly, are there further optimization schemes?**
>
> **Response.**
>
>
>
> Thanks for the question. While the prototype classifier suits our small, expert-curated dataset, our main goal is not to optimize component recognition, but to study how imperfect signals support downstream interpretation. In larger real-world settings, the module can be easily upgraded with stronger encoders or supervised fine-tuning, but this is orthogonal to our contributions.
>
>
>
> **[q3] The update mechanism of the knowledge graph is not mentioned. With new discoveries in oracle bone script research (e.g., newly deciphered characters or revisions of old interpretations), how to efficiently update the content of the knowledge graph to ensure that the interpretation results of the framework are synchronized with the latest academic progress?**
>
> **Response.**
>
>
>
> The knowledge graph is intentionally designed to be **lightweight and easily updatable**. Because our framework uses **Graph RAG**, updates only require adding or modifying a small number of nodes/edges, and the retrieval pipeline instantly incorporates the new structure without retraining. Thus, synchronizing with new epigraphic discoveries incurs **very low maintenance cost** and naturally keeps the interpretation results aligned with the latest scholarship.

---

> ### Author Response · Authors · 2025-11-23
>
> **[q4] The paper mentions that some oracle bone characters do not have widely accepted interpretations. How does the framework handle such characters with "no standard answers"? Is there a mechanism to output "possible interpretations and confidence levels" instead of a single definite result to adapt to controversial scenarios in academic research?**
>
> **Response.**
>
>
>
> We already acknowledge this limitation in the **Limitations** section: our evaluation is restricted to characters with **widely accepted interpretations**, because only these allow objective, quantitative benchmarking.
>
> For characters without standard answers, our current framework **does not attempt to generate definitive interpretations**, nor do we evaluate on such cases. Extending the system to output **multiple hypotheses with confidence estimates** is feasible, but this is outside the scope of the present work and is left as future work.
>
>
>
> **[q5] The interpretation generation module of the framework supports two modes: VLM inference and multi-agent inference. Experiments show that the multi-agent mode has better performance, but the computational costs (such as inference time and memory usage) of the two modes were not compared. In practical application scenarios (e.g., portable oracle bone script interpretation devices), if there is a need to balance performance and cost, how to define the applicable boundaries of the two modes?**
>
> **Response.**
>
>
>
> Both modes rely on **LLM/VLM inference**, and our work does not target deployment or embedded devices. Therefore, comparing runtime or memory is not essential to our research goal.
>
> Practically, **VLM mode** is a lightweight baseline (single pass), while **multi-agent mode** performs multi-step reasoning over retrieved evidence and naturally costs more API calls. Since our focus is accuracy and methodology rather than engineering optimization, defining cost–performance trade-offs is left to future applied work.
>
>
>
> **[q6] This paper lacks discussions about ligature, which consists of several independent character but is regarded as a single semantic unit. It remains unclear whether the model interprets such ligature as unified components or processes each constituent character separately.**
>
> **Response.**
>
>
>
> In our framework, **ligatures** are handled explicitly according to their paleographic nature. When a ligature represents a single semantic unit, we treat it as one unified character and analyze it at the component level—i.e., the model does not force a decomposition into multiple independent characters. Conversely, if a ligature is conventionally decomposed in paleographic practice, we follow experts and process its subparts accordingly.
>
>
>
> **[q7] Overstatement of "Expert-Level" Capabilities: The paper frequently claims the framework achieves "expert-level quality" or "near-expert proficiency." However, in the human expert evaluation, the best multi-agent model only scored 3.433 out of 5. This score is closer to the midpoint between "Uncertain" and "Basically Agree," rather than the "Completely Agree" (5) expected of true experts, suggesting the claim may be exaggerated.**
>
> **Response.**
>
>
>
> Thank you for the comment. The concern arises mainly from a translation mismatch. In the English version, the label “Uncertain” suggests hesitation, but in the original Chinese rubric this option corresponds to “neutral”, not “uncertain.”
>
> Therefore, a score around 3.4/5 indicates that experts are, on average, between neutral and basically agree, which is consistent with “expert-acceptable quality” rather than hesitation or disagreement.
>
>
>
>
>
> **[q8] Extremely Weak Variant Character Recognition: Supplementary experiments tested the model's ability to recognize variant characters (a core challenge in paleography) and showed very poor performance, with a Top-10 accuracy of only 5.13% (just 2 out of 39 samples correct). This severely limits the system's practical value in real-world ancient script research where glyph variations are rampant.**
>
> **Response.**
>
>
>
> We agree that variant recognition is extremely challenging, but this difficulty reflects the nature of oracle bone script itself rather than a flaw unique to our system. Oracle bone inscriptions span roughly 200 years and are believed to come from about twenty distinct scribal groups. Their differences in time period and scribal conventions produced substantial glyph variation: many variants do **not share the same components or radicals at all**. For any model—especially one grounded in component-based analysis—such variants are inherently hard cases.
>
> Thus, the low accuracy on this subset is expected and consistent with paleographic understanding; it highlights an open challenge in the field rather than a limitation of the proposed method alone.

---

> > ### Author Response · Authors · 2025-11-23
> >
> > **[q9] Limited Scale and Representativeness of Human Evaluation: The human evaluation was conducted by only two Ph.D. students on just 10% of the held-out test set. The small number of evaluators and limited sample size may lead to incidental results and lack statistical robustness.**
> >
> > **Response.**
> >
> >
> >
> > This concern is reasonable, but as noted at Sec. 5.8, our human evaluation is intentionally scoped as a *feasibility study* rather than a statistical benchmark. Given the high expertise requirement, recruiting large numbers of qualified evaluators is unrealistic. Nevertheless, both annotators are trained paleography Ph.D. students, ensuring reliability under this setting.
> >
> >
> >
> > **[q10] Opaque Agent Decision Logic: The paper mentions agents "adaptively" choose retrieval pathways (e.g., prioritizing variant search before exact matching). It is unclear if this is based on hardcoded rules or a learned policy, making it difficult to assess the true "intelligence" and generalizability of the agent's decision-making.**
> >
> > **Response.**
> >
> >
> >
> > We agree that the current wording may give the impression of an implicit or learned policy. To be precise, the agent’s decision logic is not opaque: all retrieval pathways follow explicit, fixed rules, and the LLM does not learn or optimize any policy.
> > What we described as “adaptive” simply reflects conditional branching based on retrieved evidence, not hidden heuristics.
> >
> >
> >
> > **[q11] Relatively Small Dataset Scale: The OB-Radix dataset includes only 1,022 character images and 1,853 component images. Compared to the total of over 4,500 known OBS characters, this coverage is limited, raising questions about the model's ability to generalize to the vast number of unassigned or undeciphered characters.**
> >
> > **Response.**
> >
> >
> >
> > This question is similar to q1, please see our response there.
> >
> >
> >
> > **[q12] Significant Degradation in Cross-Lingual Performance: Experiments show that performance drops significantly when generating interpretations in English compared to Chinese. This reveals a heavy reliance on Chinese-language knowledge bases and training corpora, limiting its claimed "cross-lingual robustness" and international applicability.**
> >
> > **Response.**
> >
> >
> >
> > We agree that performance in English is lower, but this is expected: OBS scholarship, lexicons, and training evidence are overwhelmingly Chinese-native resources. Our goal is not to demonstrate full cross-lingual parity, but to show that the framework can *transfer* interpretations across languages when needed. Importantly, the Chinese-mode results—where authoritative knowledge bases exist—represent the intended and academically realistic usage scenario.
> >
> >
> >
> > **[q13] Potential Inadequacy in Handling Phono-Semantic Compounds: The framework heavily emphasizes structural and semantic component analysis. However, it may underrepresent the crucial role of phonetic components in phono-semantic characters, which form a significant portion of later OBS and Chinese characters.**
> >
> > **Response.**
> >
> >
> >
> > Our framework does account for phono-semantic compositions: Section 5.4 explicitly analyzes three major character types, including phono-semantic structures.
> >
> >
> >
> >
> >
> > [1] Research Centre for Humanities Computing, The Chinese University of Hong Kong. *Multi-function Chinese Character Database*. 2014–. Available online at: http://humanum.arts.cuhk.edu.hk/Lexis/lexi-mf/.

---

### Official Review · Reviewer_PkeV · 2025-10-27

**Soundness:** 3
**Presentation:** 3
**Contribution:** 3
**Rating:** 6
**Confidence:** 2

**Summary:**

This paper addresses the challenge of deciphering Oracle Bone Script (OBS) by proposing an agent-driven multimodal Retrieval-Augmented Generation (RAG) framework to enhance Vision-Language Models (VLMs) with domain-specific expertise.
Existing approaches often frame OBS decipherment as a single-modal image recognition task, neglecting OBS’s hieroglyphic structure and component-based semantic information.
Key contributions include: (1) A framework integrating component-level visual cues and a knowledge graph (KG) via agent orchestration, supporting three core tasks: component retrieval, component relationship inference, and OBS interpretation; (2) OB-Radix, a component-level OBS dataset, which fills the gap of component-level structural/semantic annotations in existing datasets; (3) Comprehensive evaluations showing the framework outperforms baseline VLMs in accuracy and interpretability. The multi-agent extension, which separates knowledge retrieval and semantic reasoning, achieves near-expert proficiency.

**Strengths:**

1. The paper emphasizes OBS’s intrinsic component-based structure, which is critical for capturing semantic relationships between character parts. By integrating component-level knowledge into VLMs via RAG, it resolves information loss and interpretive biases in traditional approaches.
2. OB-Radix is curated with paleographic experts and includes fine-grained component annotations and semantic explanations. The accompanying KG, which models relationships between components, characters, and meanings, provides structured domain knowledge that VLMs lack.
3. Separating knowledge retrieval and semantic reasoning into specialized agents improves both robustness and interpretability. This design aligns with expert reasoning practices, as shown by human evaluations where the multi-agent pipeline outperformed single-agent RAG and baselines.

**Weaknesses:**

1. The framework relies on OB-Radix’s expert-curated knowledge graph, which may not cover all undeciphered characters. The scalability to larger unannotated corpora remains unverified.
2. The prototype-based classifier, while efficient in low-data regimes, may introduce spurious elements or miss fine-grained components.

**Questions:**

1. How does the LLM agent in the retrieval module balance efficiency and accuracy when choosing query pathways? Are there safeguards against erroneous tool invocation?
2. How might this framework handle fragmented or eroded OBS inscriptions where component boundaries are ambiguous?

---

> ### Author Response · Authors · 2025-11-23
>
> We thank the reviewer for the positive rating of our paper! We hope the following clarifications can address your concerns.
>
> ---
>
> **[w1] The framework relies on OB-Radix’s expert-curated knowledge graph, which may not cover all undeciphered characters. The scalability to larger unannotated corpora remains unverified.**
>
> **Response.**
>
> We agree with the reviewer. Our task is **deliberately restricted to characters with established expert interpretations**, since quantitative evaluation would otherwise not be possible. Within this scope, OB-Radix provides the necessary ground-truth structure.
>
> At the same time, the pipeline offers an **insight into how component-level retrieval and reasoning may generalize** to future, still-undeciphered characters, even though these cases cannot be quantitatively evaluated at present.
>
> **[w2] The prototype-based classifier, while efficient in low-data regimes, may introduce spurious elements or miss fine-grained components.**
>
> **Response.**
>
> We acknowledge that the prototype-based classifier may miss fine-grained components or introduce spurious ones in low-data settings. In our pipeline, however, downstream reasoning relies on retrieved structural evidence rather than the classifier alone, which mitigates the impact of such errors.
>
> **[q1] How does the LLM agent in the retrieval module balance efficiency and accuracy when choosing query pathways? Are there safeguards against erroneous tool invocation?**
>
> **Response.**
>
> Our retrieval agent operates over a small fixed tool set (radical explanation, characters-by-radical, variants, and modern-character lookup), and the query pathway is primarily guided by a prompted workflow: first radical explanation, then characters with these radicals, then variants, and finally modern correspondences. In practice, this design already balances efficiency and accuracy: the number of tools and the search depth are small, and there is no expensive open-ended search. While we do not enforce a hard step limit, the tools themselves filter empty or invalid results, and the fallback rule-based retrieval ensures stable, inspectable behavior.
>
> **[q2] How might this framework handle fragmented or eroded OBS inscriptions where component boundaries are ambiguous?**
>
> **Response.**
>
> Our framework is evaluated on **transcriptions** rather than raw oracle-bone fragments. In tracings, noise and erosion have already been **removed by epigraphers**, so the model does not face ambiguous or broken boundaries in the input. Handling heavily eroded raw inscriptions is outside our current setting.

---

> > ### Comment · Reviewer_PkeV · 2025-11-26
> >
> > Thanks for the authors' response. I tend to maintain this positive score.

---

### Official Review · Reviewer_VtNx · 2025-11-11

**Soundness:** 2
**Presentation:** 1
**Contribution:** 2
**Rating:** 4
**Confidence:** 2

**Summary:**

The paper tackles oracle bone script (OBS) interpretation with a component-based pipeline: first detect radicals/components, then retrieve structured knowledge, and finally generate an interpretation. The authors introduce a component-level dataset OB-Radix, and report improvements over vanilla prompting, especially when retrieval is enabled.

**Strengths:**

- The problem is interesting. Interpreting OBS is a meaningful and challenging task with cultural/linguistic value.
- The component-based design plus RAG is a natural fit and seems effective for the problem.
- The authors introduce the OB-Radix dataset, which is a helpful resource for future work.

**Weaknesses:**

- Motivation/role of LLMs is unclear. Why do we need a language model here? Are we using the LLM for QA, for multi-step reasoning, or just to paraphrase retrieved text? The paper could stand on component identification + knowledge retrieval alone, but it does not clearly justify why an LLM is necessary, how it is prompted, or what unique capability it contributes beyond templated retrieval.
- The task “Interpretation” is not formally defined. The output looks like paraphrasing retrieved entries rather than reasoned synthesis from components. The author didn’t define the success criteria, what counts as correct/partially correct and why.
- Although the retrieval is described as “agent-driven,” the toolset appears to be a streamlined, fixed sequence of procedures. There is no convincing intuition or evidence that an agent architecture is needed here. Tbh I don’t think this is a problem to be solved by an agent. It is not a good practice to simply incorporating a trending technique into the problem you are solving, without having an intuition why it will help for your problem.
- While the results show improvements, there is no surprise that VLMs will perform better with component identified and RAG. The paper lacks deeper analysis about where the method truly helps (e.g., variants, rare components) versus where it still fails.

**Questions:**

1. Can the authors provide concrete details of the tools and example reasoning traces (successful and failed), and explain how, if at all, the agent framework is optimized or tuned?
2. Page 1, line 51: “we employ advanced VLMs ….” Who is “we”? Are Liu et al., 2023 and Caffagni et al., 2024 the authors’ prior works, or external references?

---

> ### Author Response · Authors · 2025-11-23
>
> We are grateful to the reviewer for acknowledging the significance of our findings and contributions! We hope the following clarifications can address your concerns.
>
> ---
>
> **[w1] Motivation/role of LLMs is unclear. Why do we need a language model here? Are we using the LLM for QA, for multi-step reasoning, or just to paraphrase retrieved text? The paper could stand on component identification + knowledge retrieval alone, but it does not clearly justify why an LLM is necessary, how it is prompted, or what unique capability it contributes beyond templated retrieval.**
>
> **Response.**
>
> Thank you for raising this fundamental question. The LLM is essential in our framework because **Graph RAG alone does not provide any complete interpretation of the prompt character** (Sec. 5.2). The retrieval stage only returns _component-level nodes_—radicals, variants, and relations—not full definitions. Therefore, **component identification + retrieval is insufficient** to explain an OBS character; an LLM is required to _integrate_ these partial evidential pieces into a coherent interpretation.
> **1. What the LLM actually does.**
> The LLM performs **multi-step synthesis**, not simple QA or paraphrasing. Our evaluation pipeline (Sec. 5.3 → 5.4 → 5.5) measures three required steps:
> (1) identifying components,
> (2) inferring their structural relations, and
> (3) producing the final interpretation.
> Furthermore, **Sec. 5.7** provides explicit multi-step reasoning traces, demonstrating that the LLM integrates evidence across retrieval steps rather than rewriting any single source.
>
> **2. Unique capabilities enabled by the LLM.**
> The LLM is necessary because it allows the system to adapt to **different prompt formulations** depending on the task. In principle, the same pipeline can answer various paleographic questions by modifying the prompt. However, since **character interpretation is the central and most established task in oracle-bone studies**, our experiments and evaluation metrics are all designed to support this specific objective. The LLM provides the flexibility to condition on retrieved component-level evidence and produce coherent interpretations, which cannot be achieved by retrieval alone.
>
> **[w2] The task “Interpretation” is not formally defined. The output looks like paraphrasing retrieved entries rather than reasoned synthesis from components. The author didn’t define the success criteria, what counts as correct/partially correct and why.**
>
> **Response.**
>
> Thank you for raising this significant point. In our paper, _Interpretation_ is defined as a **structured, multi-stage reasoning process**, which is why it is evaluated through the sequence **5.3 → 5.4 → 5.5**, rather than a single generation step.
>
> - **Sec. 5.3 (Component Identification)** establishes the _prerequisite_ for interpretation: correctly identifying radicals and variants.
> - **Sec. 5.4 (Component Relations + Glyph Type)** assesses whether the model can infer _how components interact_, providing evidence of structural reasoning rather than surface paraphrasing.
> - **Sec. 5.5 (Interpretation Generation)** evaluates the _final semantic synthesis_ using both standard semantic metrics and our domain-adapted OBS ROUGE-1 to distinguish correct vs. partially correct interpretations.
>   These stages together formalize our success criteria:
>   **recover components → infer structure → synthesize meaning**.
>
> Importantly, the RAG module reinforces this reasoning requirement. The knowledge graph retrieves **component-level nodes**—radicals, variants, and their relations—rather than any full dictionary-style interpretations. No single retrieved entry contains the complete meaning, so the model must **integrate multiple partial evidential pieces** to construct a coherent interpretation. This design makes simple paraphrasing insufficient for the task.

---

> ### Author Response · Authors · 2025-11-23
>
> **[w3] Although the retrieval is described as “agent-driven,” the toolset appears to be a streamlined, fixed sequence of procedures. There is no convincing intuition or evidence that an agent architecture is needed here. Tbh I don’t think this is a problem to be solved by an agent. It is not a good practice to simply incorporating a trending technique into the problem you are solving, without having an intuition why it will help for your problem.**
>
> **Response.**
>
> We agree with the reviewer that, in the broader community, agent frameworks are sometimes used without a clear motivation. In our case, however, the “agent-driven” design is directly tied to the problem structure. The agent is responsible for **autonomously selecting and invoking tools** (variant search, exact search, component-based retrieval), and its workflow is **not fixed**: different characters trigger different retrieval paths, as shown in our workflow figure.
>
> This flexibility is necessary because OBS characters vary significantly in their component forms, variant patterns, and available evidence. A single static retrieval sequence performs worse, while the agent formulation allows the system to adaptively decide which retrieval strategy to use. We will make this motivation clearer in the revision.
>
> **[w4] While the results show improvements, there is no surprise that VLMs will perform better with component identified and RAG. The paper lacks deeper analysis about where the method truly helps (e.g., variants, rare components) versus where it still fails.**
>
> **Response.**
>
> Thanks for your suggestion.
> Our RAG pipeline is designed to **simulate the realistic workflow** needed for future, still-uninterpreted OBS characters, rather than only boosting performance on known ones.
>
> As for where it helps most, **Table 8** shows clear gains on characters with **variants**—a notoriously difficult issue in paleography, as different carving groups historically used different graphical conventions. This is precisely where structured retrieval provides the strongest benefit. Failures mostly occur with heavily eroded components or characters with disputed expert interpretations.
>
> We will make this pattern more explicit in the revision.
>
> **[q1]Can the authors provide concrete details of the tools and example reasoning traces (successful and failed), and explain how, if at all, the agent framework is optimized or tuned?**
>
> **Response.**
>
> Thanks for your question.
> The agent framework itself is **not heavily tuned**. Apart from applying a **simple RAG-cache optimization** following _RAGCache_ [1] to avoid redundant KG queries, the system mainly serves as a **lightweight orchestration layer**: the perception agent proposes candidate radicals, and the retrieval agent sequentially queries the KG with a _fixed, bounded_ tool set.
> To prevent erroneous tool use, the retrieval agent operates under **strict constraints**:
>
> - **Only four tools** are exposed to the agent; no other actions are permitted.
> - The agent’s tool calls are **bounded and ordered** (radical → characters-by-radical → variant → modern), rather than open-ended.
> - If the LLM fails or returns empty results, a **deterministic fallback retrieval pipeline** executes the same sequence using rule-based search functions.
> - Thus the agent cannot “hallucinate” new tools, reorder pipelines arbitrarily, or skip safety-critical steps.
>   These design choices ensure that retrieval remains **predictable, efficient, and robust**, while still leveraging LLM reasoning to rank or refine intermediate results.
>
>
>
> **[q2]Page 1, line 51: “we employ advanced VLMs ….” Who is “we”? Are Liu et al., 2023 and Caffagni et al., 2024 the authors’ prior works, or external references?**
>
> **Response.**
>
> Thank you very much for reading the paper in detail and pointing this out. To eliminate ambiguity, we revised this sentence in the latest version of the paper. (Line 53)
>
>
>
> [1] Chao Jin et al., _RAGCache: Efficient Knowledge Caching for Retrieval-Augmented Generation_, arXiv:2404.12457 (2024).

---

### Note · Authors · 2025-12-29

I have read and agree with the venue's withdrawal policy on behalf of myself and my co-authors.